# ON THE COMPUTATIONAL INEFFICIENCY OF LARGE BATCH SIZES FOR STOCHASTIC GRADIENT DESCENT

## ABSTRACT

Increasing the mini-batch size for stochastic gradient descent offers significant opportunities to reduce wall-clock training time, but there are a variety of theoretical and systems challenges that impede the widespread success of this technique (Das et al., 2016; Keskar et al., 2016). We investigate these issues, with an emphasis on time to convergence and total computational cost, through an extensive empirical analysis of network training across several architectures and problem domains, including image classification, image segmentation, and language modeling. Although it is common practice to increase the batch size in order to fully exploit available computational resources, we find a substantially more nuanced picture. Our main finding is that across a wide range of network architectures and problem domains, increasing the batch size beyond a certain point yields no decrease in wall-clock time to convergence for *either* train or test loss. This batch size is usually substantially below the capacity of current systems. We show that popular training strategies for large batch size optimization begin to fail before we can populate all available compute resources, and we show that the point at which these methods break down depends more on attributes like model architecture and data complexity than it does directly on the size of the dataset.

## 1 INTRODUCTION

Mini-batch stochastic gradient descent (SGD) is the dominant optimization method for training deep neural networks (DNNs) (Bengio & LeCun, 2007; Bottou, 2010). In the face of unprecedented growth in dataset size, a large body of work has attempted to scale SGD to train DNN models on increasingly large datasets, while keeping *wall-clock time* manageable (Iandola et al., 2015; Goyal et al., 2017; Smith & Le, 2018; Devarakonda et al., 2017). The most common approach to train large models at scale is distributed synchronous mini-batch SGD, which exploits additional computational resources through data parallelism. This technique reduces wall-clock training time by increasing the mini-batch size, i.e., the number of examples used to compute a stochastic estimate of the gradient of the loss function at each training iteration, while holding the number of epochs constant. Proponents of large batch size training often argue that the merits stem from its ability to decrease wall-clock training time while maintaining final model performance. Indeed, an enormous amount of work has gone into designing systems that seem to operate under an assumption that equates large batch size training with machine learning at scale (Goyal et al., 2017; Jia et al., 2018; Puri et al., 2018).

Increasing the batch size improves the scaling performance of SGD per epoch, and there are significant challenges in building efficient distributed systems that are able to exploit additional computational resources to use large batch sizes (Jia et al., 2018). However, even if we were able to address these systems challenges, there are still more fundamental limitations to this approach. Large batch sizes often negatively impact important performance metrics of interest, including total computational cost (which usually determines monetary cost) and prediction quality.

In this paper, we will measure the total computational cost as the number of training iterations times the work done per iteration—in order to simplify measurements, we use the number of training iterations as a proxy for the wall-clock time. We do this because the implementation of parallel algorithms depends on software and hardware choices, and our goal is to draw more general conclusions about the performance of SGD-based methods.

Based on this model for total computational cost and wall-clock time, the following should be clear: unless increasing the batch size leads to a commensurate decrease in the total number of training iterations *needed to find a good model*, large batch training will result in greater total computational cost with little-to-no decrease in wall-clock training time.

Based on our empirical results across a range of datasets and architectures, we find that as the batch size becomes larger, there are three main phases of scaling behavior for convergence speed: (1) there is a small regime of batch sizes in which increasing the batch size results in linear gains in convergence speed; (2) there is a larger regime of batch sizes that results in sublinear gains in convergence speed—in this regime, increasing the batch size can improve wall-clock training time at the expense of greater total computational cost; (3) eventually, we reach a third regime where a higher batch size results in marginal or non-existent reductions in convergence speed. In our experiments, we find that this third regime begins at a batch size that is too small to let us fully utilize available compute. *Training past this batch size increases the total computational cost without reducing wall-clock training time or prediction quality.* While there has been considerable excitement around heuristics that have been shown to make large batch training practical for certain problems (Goyal et al., 2017; Smith & Le, 2018), we demonstrate that these techniques still suffer from the same convergence trends we observe, and they often decrease stability of the training process.

Recent work has observed that the final *test performance* of models trained with large batch sizes degrades after training for a fixed number of epochs (Yao et al., 2018; Keskar et al., 2016). This phenomenon is known as the *generalization gap*. Previous work addressing this problem has focused on training for more iterations in the large batch case (Hoffer et al., 2017) or adopting various heuristics to select a learning rate for larger batch sizes (Goyal et al., 2017; Smith & Le, 2018). Based on our empirical results, we find that existing techniques to mitigate the generalization gap do not work on some problems, and for other problems they only work for batch sizes that do not allow us to fully utilize our available compute. Perhaps more importantly, they do little to affect the diminishing returns in rates of convergence for training loss as batch size increases.

Our objective is to understand the behavior of SGD and existing large batch techniques for many network architectures and problem domains, e.g., image classification/segmentation and natural language processing (NLP). We observe markedly worse performance for these techniques in domains other than image classification, where large batch optimization has received the most attention (Jia et al., 2018; You et al., 2017b). Because we eschew the challenges of an efficient distributed implementation by measuring number of iterations instead of wall-clock time, our results assume the most optimistic circumstances for large batch training. Our key observations are:

- **Increasing the batch size beyond a certain point yields no improvement in wall-clock time to convergence, even for a system with perfect parallelism.** We observe that larger batch sizes result in a limited reduction in the number of training iterations needed to achieve low training or test error, and that eventually these gains become near-zero.

- **Increasing the batch size leads to a significant increase in generalization error, which cannot be mitigated by existing techniques.** We observe that these techniques often result in divergent training behavior or that they only mitigate degradation in test performance for small batch sizes relative to available compute.

- **Dataset size is not the only factor determining the computational efficiency of large batch training.** We observe that both the diminishing returns in convergence speed and the failure of existing methods correlate more with factors like model architecture and data complexity than dataset size alone. As a result, training time may significantly increase with dataset size in spite of increasingly available compute resources.

In Section 2, we review the formulation of SGD as well as existing strategies to train with large batch sizes. In Section 3, we review recent theoretical results regarding the convergence rates of SGD in highly over-parameterized settings and discuss the potential impact of these results on the computational efficiency of SGD for deep learning. Section 4 presents our empirical results that demonstrate the inefficiencies of training SGD with large batch sizes, and we show that these persist when using existing large batch optimization techniques.

## 2 BACKGROUND AND RELATED WORK

**Stochastic Gradient Descent.** SGD is the most widely used algorithm to train DNN models. The model is parameterized by weights $\mathbf{w} \in \mathbb{R}^d$, and the objective is to minimize the empirical loss over $n$ data points $\mathbf{x}_i$:

$$L(\mathbf{w}) = \frac{1}{n} \sum_{i=1}^{n} \ell(\mathbf{w}, \mathbf{x}_i), \tag{1}$$

where $\ell(\cdot, \cdot)$ is a loss, e.g., cross-entropy or squared error. This loss gives a corresponding gradient

$$\mathbf{g}(\mathbf{w}) := \nabla L(\mathbf{w}) = \frac{1}{n} \sum_{i=1}^{n} \nabla \ell(\mathbf{w}, \mathbf{x}_i). \tag{2}$$

A mini-batch $\mathcal{B}_m$ of size $m < n$ is a collection of $m$ indices randomly drawn from the set $\{1, \ldots, n\}$, and we can use it to form an unbiased estimate of the gradient at iteration $k$, as well as the corresponding SGD update:

$$\mathbf{g}_m(\mathbf{w}_k) = \frac{1}{m} \sum_{i \in \mathcal{B}_m} \nabla \ell(\mathbf{w}_k, \mathbf{x}_i) \quad \text{and} \quad \mathbf{w}_{k+1} = \mathbf{w}_k - \eta_k \, \mathbf{g}_m(\mathbf{w}_k), \tag{3}$$

where $\eta_k > 0$ is the learning rate for iteration $k$. One iteration of training for SGD corresponds to a single gradient computation / weight update. One epoch corresponds to $n/m$ iterations of training. This constitutes a single pass over the dataset, assuming the dataset is sampled without replacement.

Efficient distributed systems reduce wall-clock training time by parallelizing gradient calculations across many machines. When the batch size is large enough to populate all available compute resources, this allows us to amortize the cost of coordination for each weight update.

**Existing large batch techniques.** With the hope of keeping training times manageable as dataset sizes escalate, recent work has focused on the development of techniques that allow practitioners to increase the batch size to make use of growing computational resources (Jin et al., 2016; Jia et al., 2018; You et al., 2017a). However, there is a growing body of theoretical and empirical results suggesting that large batch sizes adversely affect the generalization performance of the final model (Yao et al., 2018; Keskar et al., 2016; Devarakonda et al., 2017).

In response to this, recent work has proposed changing two parameters in relation to batch size: the number of training iterations and the learning rate. However, they also make assumptions that limit the effectiveness of their proposals as useful heuristics for practitioners.

- *Training longer:* Hoffer et al. (2017) suggest increasing the number of training iterations. Even if this does reduce the generalization gap, it significantly increases both wall-clock training time and computational cost. Moreover, in some problems it does not lead to minima with better generalization performance (as we found when running our experiments).

- *Square root LR scaling:* Scaling the learning rate as $\eta_0 \propto \sqrt{m}$ attempts to keep the weight increment length statistics constant, but the distance between SGD iterates is governed more by properties of the objective function than the ratio of learning rate to batch size (Chaudhari & Soatto, 2017; Zhu et al., 2018). This rule has also been found to be empirically sub-optimal in various problem domains (Krizhevsky, 2014).

- *Linear LR scaling:* The performance of large batch training can also be improved by using the linear scaling rule, which suggests choosing a learning rate proportional to the batch size ($\eta_0 \propto m$) (Goyal et al., 2017). There are two motivations for this rule: the first assumes that one large-batch gradient step should resemble a series of small-batch gradient steps in order for convergence rates to improve linearly (Goyal et al., 2017); the other regards the SGD update equation as the Euler-Maruyama discretization of a stochastic differential equation (Sauer, 2012; Xing et al., 2018), and attempts to maintain a constant level of mini-batch noise to help SGD explore the loss landscape (Chaudhari & Soatto, 2017; Zhu et al., 2018; Smith & Le, 2018).

Both justifications for the linear scaling rule implicitly impose strong conditions on the loss function by requiring that it behave linearly near SGD iterates; therefore, if the loss function is highly

nonlinear along the SGD trajectory or the step size is not small enough, then we should not expect these rules to provide useful guidance for many problems. Whereas several groups have successfully used this rule to train on the ImageNet dataset in under an hour, e.g. (Goyal et al., 2017; You et al., 2017b), applying this heuristic to other datasets has not led to similarly impressive results so far (Puri et al., 2018).

The focus of this paper, however, is on more fundamental limitations of large batch training, and we empirically show that the above approaches fail to prevent diminishing returns in the rate of convergence for large batch sizes. We believe that these diminishing returns are of more immediate concern than the generalization gap and warrant more careful examination: if we cannot even minimize training error quickly, there is no real opportunity to minimize test error quickly, regardless of the difference in final test error across batch sizes by the time the model has converged.

## 3   CRITICAL BATCH SIZES AND DIMINISHING RETURNS

The convergence rate of SGD, denoted by $k_\epsilon(m)$, is the number of iterations needed to achieve training error less than a fixed constant $\epsilon > 0$ by using SGD with batch size $m$ (we will drop the subscript $\epsilon$ when it is unambiguous). In order to guarantee that large batch sizes speed up training, $k(m)$ should continue to decrease near-linearly with $m$. Otherwise, a larger batch size increases computational cost with only limited reductions in wall-clock training time. For near-constant $k(m)$, the benefit of large batch sizes becomes near-zero.

Ma et al. (2017) showed theoretically that in convex, over-parameterized settings, the reduction in convergence time obtained by increasing the batch size decays dramatically to a near-constant level after a *critical batch size* that is independent of the dataset size. This speedup is measured with respect to the number of SGD iterations required to reach some fixed loss error for some baseline batch size $m_0$, and for this purpose we define the *speedup ratio* $s(m; m_0) = k(m_0)/k(m)$. The speedup ratio represents the amount of time we save by increasing the batch size to $m$. Beyond the critical batch size mentioned above, even with no communication overhead and unlimited resources (where each batch size requires the same amount of wall-clock time to process) we would prefer to use the critical batch size because it requires less overall computation.

This result is surprising because researchers have asserted that it should be possible to achieve linear gains in convergence speed so long as the batch size is small relative to dataset size (Smith & Le, 2018). This will present significant difficulties for future optimization work (large mini-batch training) because it prevents us from using large batch sizes as a catch-all approach to quickly train models on large datasets.

## 4   EMPIRICAL EVALUATION

Recent work studying large batch training has looked primarily at image classification (Jastrzebski et al., 2018; Yao et al., 2018), especially on the ImageNet dataset (Deng et al., 2009). We perform large batch size experiments across both traditional image classification (IC) tasks (such as on CIFAR-10/100 (Krizhevsky & Hinton, 2009)), as well as previously unexplored tasks like image segmentation (IS) using the Cityscapes dataset (Cordts et al., 2016), and natural language processing (NLP) using the WikiText-2 dataset (Merity et al., 2016). We also test how these results vary across other modern DNN architectures, namely ResNets (He et al., 2016), LSTMs (Hochreiter & Schmidhuber, 1997; Gers et al., 2000), AlexNet (Krizhevsky et al., 2012), VGG (Simonyan & Zisserman, 2014), Dilated Residual Networks (Yu et al., 2017), and MobileNetV2 (Sandler et al., 2014). We tested all of the large batch training techniques described in Section 2. We tried training longer based on the work of Hoffer et al. (2017), but we found that this necessarily cannot improve the convergence speed and often does not improve final test performance. The two other techniques include the square root scaling rule strategy (SRSR) and the linear scaling rule strategy (LSR). For the latter, we used a warm-up period at the start of training as suggested by Goyal et al. (2017). Table 1 reports our datasets, models and different training strategies. For each model, we evaluated against a base learning rate strategy (BLR) that used the same learning rate across all batch sizes. We selected this learning rate based on its performance on a small baseline batch size.

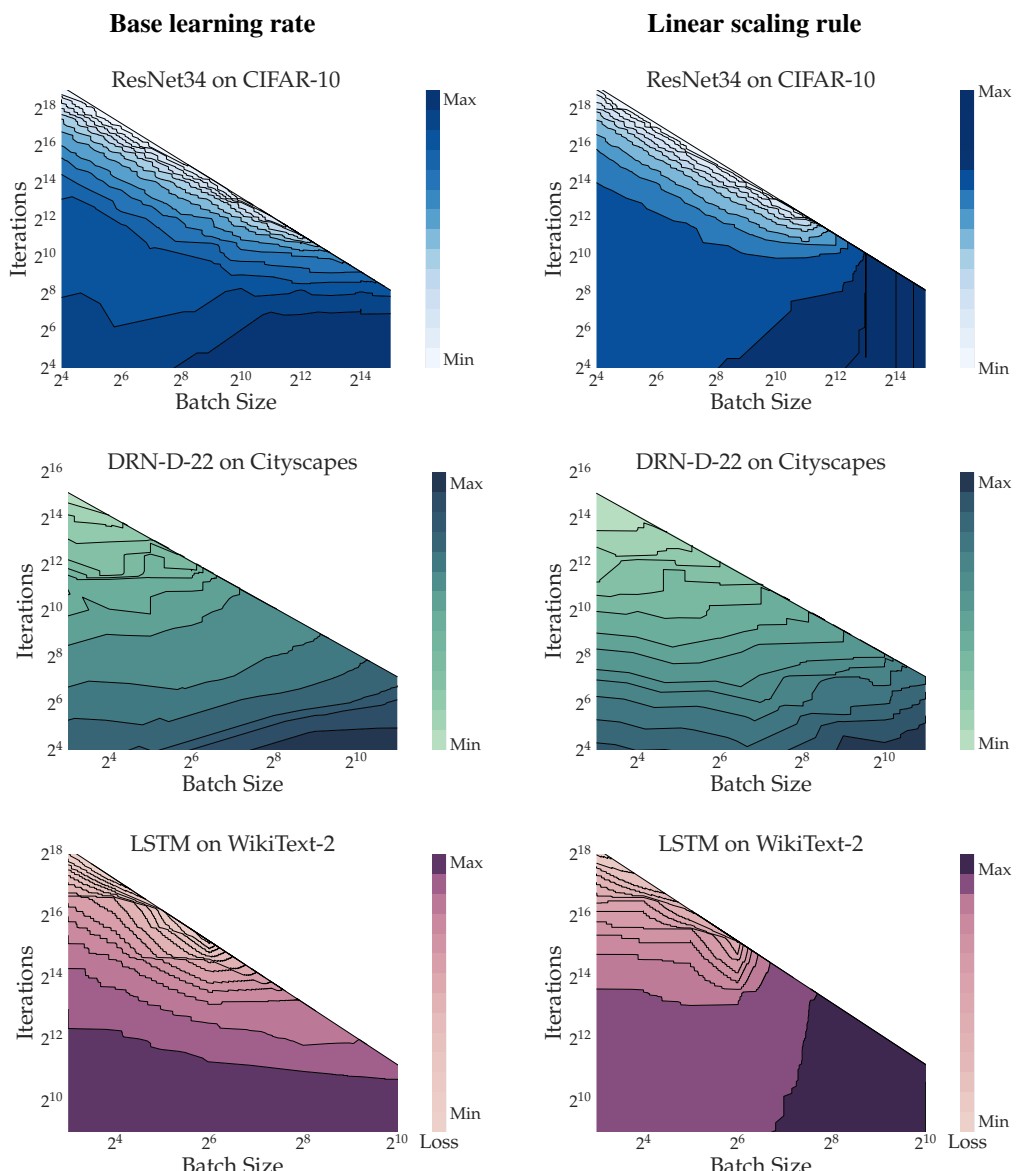

Figure 1: **Contour plots of training losses** for various problem domains on a log scale. Lighter colors indicate lower loss values. Since we train each batch size for a fixed number of epochs, the total number of training iterations scales down linearly. For each loss value, we can observe how many iterations it takes to converge to that value given a particular batch size, by tracing the level curve for the associated color. For all problems, there is a batch size after which the number of training iterations necessary to converge does not decrease.

## 4.1 DIMINISHING RETURNS IN RATES OF CONVERGENCE

We demonstrate the rapidly diminishing returns in rates of convergence across various problem domains and network configurations. Researchers increase the batch size in an attempt to achieve nearly linear speedups in convergence compared to a small mini-batch size. In particular, if the speedup is near-linear, i.e. $s(m; m_0) = k(m_0)/k(m) \approx m/m_0$, then the computational cost remains nearly constant for large and small mini-batch SGD. However, if $s(m) \ll m/m_0$, then the benefit of using large batch size training is negligible.

In Figure 1, we show contour plots of training loss as a function of both the batch size and the number of training iterations of ResNet34 on CIFAR-10, an LSTM on WikiText-2, and DRN-D-22

Table 1: A description of the problem configurations and training strategies used in this paper. $\eta_0$ is the initial learning rate, $W$ is the number of epochs used for warm-up in the linear scaling rule, $E$ is the total number of epochs trained

| Dataset | Task | Architecture | Training Strategy | BS range |
|---|---|---|---|---|
| MNIST | IC | ResNet34 | BLR, LSR ($\eta_0 = 0.1, W = 10, E = 200$) | $2^6 - 2^{14}$ |
| CIFAR-10 | IC | AlexNet, MobileNetV2 ResNet34, VGG16 | BLR, LSR, SRSR ($\eta_0 = 0.1, W = 10, E = 200$) | $2^6 - 2^{14}$ |
| CIFAR-100 | IC | ResNet34 | BLR, LSR ($\eta_0 = 0.1, W = 10, E = 200$) | $2^6 - 2^{14}$ |
| SVHN | IC | ResNet34 | BLR, LSR ($\eta_0 = 0.1, W = 10, E = 200$) | $2^6 - 2^{14}$ |
| WikiText-2 | NLP | LSTM | BLR, LSR ($\eta_0 = 20, W = 3, E = 40$) | $2^3 - 2^{10}$ |
| Cityscapes | IS | DRN-D-22 | BLR, LSR ($\eta_0 = 0.01, W = 10, E = 100$) | $2^3 - 2^{11}$ |

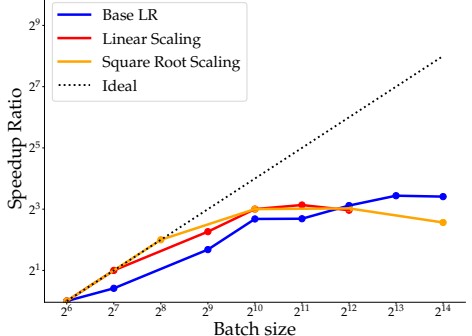 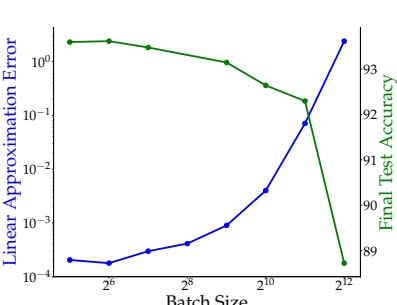

Figure 2: On the left: speedup curves when applying several popular techniques to avoid the generalization gap. Base LR uses a single learning rate for all batch sizes. On the right: the effect of the linear approximation error on final test accuracy when using the linear LR scaling rule.

on Cityscapes. Consider, for example, the contour plot for ResNet34 trained on CIFAR-10. We can see that as the batch size increases from 16 to 2048, the number of SGD iterations needed to achieve a particular loss value decreases linearly. Exceeding this regime, however, the speedup ratio becomes increasingly sublinear and soon we have $s(m; m_0) \ll m/m_0$. For batch size 8196, the training procedure does not achieve the lowest training loss, and from this perspective, even if we did not care about computational cost or training time, we would not be able to find an accurate model. We observe even worse scaling behavior for test performance (please see Figure 5 for details).

For NLP and IS, note that the gain from large batch training diminishes even faster. Neither the LSTM on WikiText-2 nor DRN-D-22 on Cityscapes can reach their respective baseline performances after reasonably small batch sizes of about 32 and 64, respectively. Although Puri et al. (2018) showed that training on the Amazon Reviews dataset (McAuley et al., 2015) can be done within 4 hours, they tune hyper-parameters heavily. This poses an issue for many practical deployments because these problems are often already slow to train.

## 4.2 EXISTING STRATEGIES BREAK DOWN FOR LARGE BATCH SIZES

We further explore how training with the linear and square root scaling rules compares to training with a fixed baseline learning rate (BLR) that does not change with batch size. In the left subfigure of Figure 2, we show the speedup curves of BLR, LSR, and SRSR strategies for ResNet34 on CIFAR-10. Note that LSR and SRSR outperform BLR from batch size 256 to 2048 which implies that LSR and SRSR can help the model train for small-to-medium batch sizes. However, the speedup of LSR and SRSR is still worse than the ideal linear case, and the curves plateau quickly after a batch size of 2048, at which point BLR becomes better than LSR and SRSR. This means that for certain problems, scaling up the learning rate to compensate for an increased batch size hurts performance.

In the right subfigure of Figure 2, we plot the test performance and the approximation error for LSR of ResNet34 on CIFAR-10. We measure the approximation error at the end of training, with final

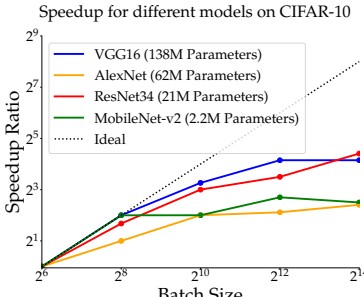 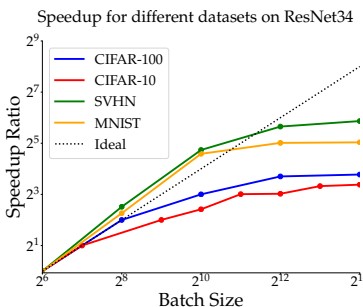

Figure 3: Speedup curves across different problem configurations. Left: different architectures result in different rates of convergence on CIFAR-10. Right: ResNet34 exhibits different rates of convergence on CIFAR-10, CIFAR-100, and SVHN. Loss thresholds are obtained by computing the lower quartile of loss values achieved by the largest batch size.

weights $\mathbf{w}^*$. We take this error to be the absolute difference between the true loss value $L(\mathbf{w})$ and the linear approximation at $\mathbf{w}^*$, given by $\hat{L}(\mathbf{w}) = L(\mathbf{w}^*) + \langle \mathbf{g}_m(\mathbf{w}^*), \mathbf{w} - \mathbf{w}^* \rangle$. The approximation is calculated for $\mathbf{w} = \mathbf{w}^* - \eta \frac{m}{m_0} \mathbf{g}_m(\mathbf{w}^*)$ to understand the behavior of the approximation along the trajectory for a single SGD iterate using the LSR. It appears that there exists a strong relationship between linear approximation error and test accuracy: as the linear approximation error increases, the test accuracy drops. Note the transition that happens at the critical batch size of 2048. After this point, the test accuracy drops significantly and the linear approximation error exceeds 1, showing that we quickly exit the regime in which the linear approximation is valid.

### 4.3 CONVERGENCE SPEED HAS A WEAK DEPENDENCE ON DATASET SIZE

Previous works have conjectured that the maximum batch size that can result in a good model is proportional to the size of the whole dataset (Smith et al., 2017; Smith & Le, 2018). However, for convex, over-parameterized problems, Ma et al. (2017) show that there is a model-dependent critical batch size after which we observe rapidly diminishing returns in convergence speed. In this section, to observe if a similar critical batch size exists in the non-convex case, we compare how changing model architecture or data complexity affects the shapes of speedup curves compared to changing the dataset size alone.

First, in order to show that these diminishing returns depend on data complexity and DNN architecture, we plot speedup curves in Figure 3 to compare the scaling behaviors across different models and dataset configurations. For the error threshold $\epsilon$, we chose the lowest quartile loss value reached by the largest batch size to make a fair comparison across configurations. This setup actually favors the large batch case, because there are lower loss thresholds that are attainable only in the small batch case. On the left, for the CIFAR-10 dataset, we compared four model architectures. For each architecture, we plotted the speedup curve obtained by training this model on the dataset for various batch sizes. The variety of speedup curve shapes indicates that model architecture is an important factor in determining the convergence speed of training for large batch sizes. For MobileNetV2/AlexNet, the diminishing returns become visible when batch size is 1024. However, for VGG16/ResNet34, the speedup does not flatten out until batch size 8196. Hence, in practice, the choice of model strongly affects our ability to use large batch sizes in SGD.

On the right, in order to investigate the effect of problem complexity, we compared the performance of ResNet34 on four datasets of the same size: CIFAR-10, CIFAR-100, MNIST, and the SVHN dataset (we cut off MNIST and SVHN to $50k$ training examples each). Although all problems display diminishing returns in rates of convergence, the point at which the curves plateau varies according to problem complexity. It is not hard to see that, for simpler problems such as SVHN, the curves flatten out later than for harder problems (e.g. CIFAR-10/100).

In all of the above cases, the diminishing rates of return in convergence speed become visible after only moderate increases in the batch size. Previous works have only studied convergence behavior for a fairly limited range of batch sizes (e.g., up to $4096$ for CIFAR-10) (Hoffer et al., 2017; Keskar

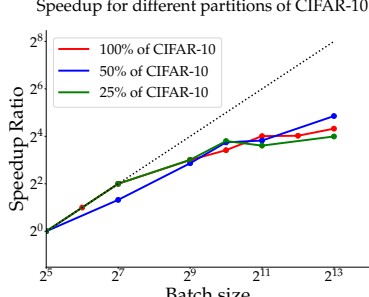 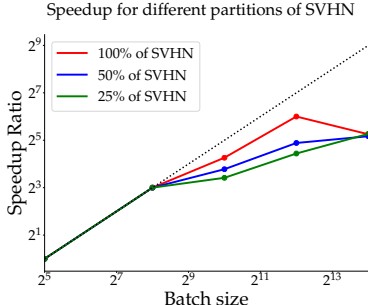

Figure 4: Speedup curves as dataset size varies for different datasets. Even as dataset size increases back up to the baseline of 100%, there is no noticeable improvement in convergence speed.

et al., 2016). By increasing the batch size past this point, it becomes immediately apparent that the primary issue with large batch size optimization is training speed, not the generalization gap.

In order to test whether the sublinear behavior of $s(m; m_0)$ depends primarily on dataset size, we compare the speedup curves obtained when training a single model on different fractions of the original training data. We trained ResNet34 models on the CIFAR-10 and SVHN datasets (for SVHN in this experiment, we train on all $600k$ available training images). For each dataset, we trained on 100%, 50%, and then 25% of the available training data.

In Figure 4, we plot the resulting speedup curves for the various partitions. In order to maintain a fair comparison (as baseline loss values change for different dataset sizes), we again choose the loss threshold to be the lower quartile of loss values obtained by the largest batch size.[1] Notably, the batch size at which the curves begin to plateau remains constant as dataset size changes. For ResNet34 on CIFAR-10, the linear speedup behavior breaks around batch size 128 for all three curves. By a batch size of 1024, all curves have flattened. We can see similar behavior for ResNet34 on SVHN. Overall, looking back to Figure 3, the choice of model and the complexity of the dataset appear to be more related to the shape of speedup curve than dataset size alone.

## 5    CONCLUSION

By experimenting across a wide range of network architectures and problem domains, we find that, after a certain point, increasing the batch size fails to decrease wall-clock time to convergence and results in low computational efficiency, even assuming perfect parallelism. The critical batch size after which these returns diminish tends to be small relative to existing system capabilities. These trends present impediments to progress in developing effective machine learning systems that are capable of handling growing data demands.

Recent works also suggest heuristics to decrease the generalization gap, but we find that these heuristics cannot be used to solve the underlying issue of training convergence speed. Moreover, we find that they usually only help decrease the generalization error in a small-to-medium batch size regime. There does not seem to be a simple training heuristic to improve large batch performance in general.

These results suggest that we should not assume that increasing the batch size for larger datasets will keep training times manageable for all problems. Even though it is a natural form of data parallelism for large-scale optimization, alternative forms of parallelism should be explored to utilize all of our data more efficiently.

---

[1]We observed that the loss threshold for a smaller partition is higher than that of the full dataset. This may be because, as we decrease dataset size, the large batch behavior that determines our threshold approaches that of vanilla gradient descent, which typically displays poor training convergence speed for DNN problems.

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

# A    ADDITIONAL RESULTS

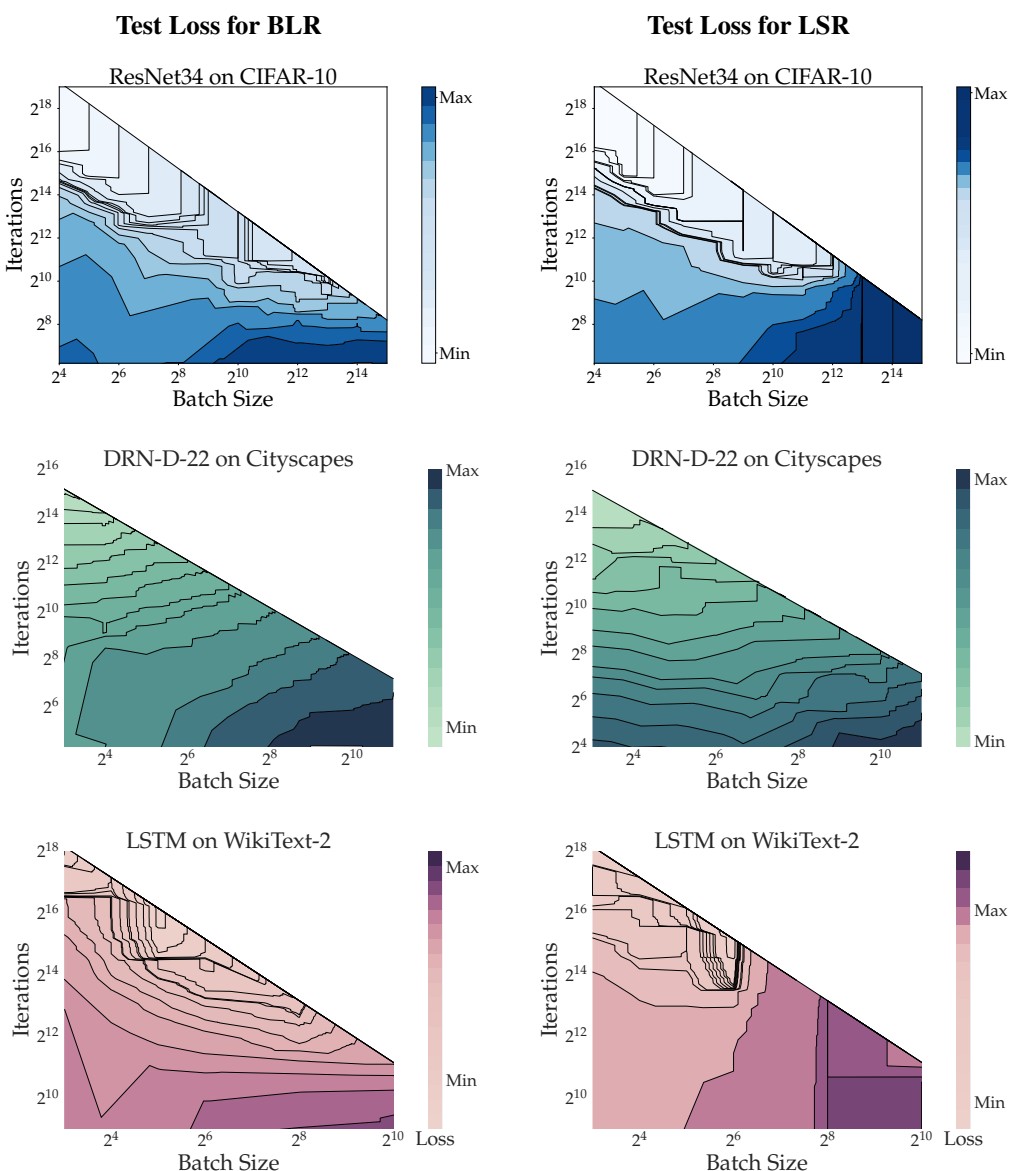

Figure 5: **Contour plots of test losses** for various problem domains on a log scale. The test losses for BLR are on the left, while the losses for the LSR strategy are on the right. Lighter colors indicate lower loss values. Since we train each batch size for a fixed number of epochs, the total number of training iterations scales down linearly. For each loss value, we can observe how many iterations it takes to converge to that value given a particular batch size, by tracing the level curve for the associated color. For all problems, there is a batch size after which the number of training iterations necessary to converge does not decrease.

