# OpenReview forum: "On the Computational Inefficiency of Large Batch Sizes for Stochastic Gradient Descent"
_ICLR.cc/2019/Conference_

### Official Review · AnonReviewer2 · 2018-10-29
**Some interesting empirical results for a popular problem**

**Rating:** 5
**Confidence:** 3

**Review:**

Summary:
The authors present an empirical analysis of how the size of SGD batches affects neural networks' training time.

Strengths:
As mini-batches training is highly popular nowadays, the problem emphasized by the authors may have a high impact in the community. Together with recent analysis on the generalization properties of over-parametrized models, the paper may help understand more general open problems of neural networks' training. A nice contribution of the paper is the observation that different phases of scaling behaviour exist across a range of datasets and architectures.

Weaknesses:
Based on empirical evaluation, the paper cannot make any claim about the generality of the obtained results. Even if the authors' analysis is based on a large set of benchmarks, it is hard to asses whether and how the results extend to cases that are not included in Section 4. In particular, it is not clear how the definition of different training phases can help the practitioner to tune the training parameters, as the size and range of the different regimes depend so strongly on the model's architecture and dataset at hand.

Questions:
- have the properties of mini-batches training been explored from a formal/theoretical perspective? do those results match and confirm the proposed empirical evaluation?
- how are the empirical results obtained in the experiment section expected to depend on the specific dataset/benchmark? For example,  given a particular architecture, what are the key features that define the three training phases (shape of the nonlinearity, number of layers, underlying distribution of the dataset)?
- what is a  batch size that does not allow one to 'fully utilize our available compute'?
- does the amount of over-parameterization in the model have any effects on the definition of the training phases? How are the results obtained in the paper linked to the generalization gap phenomenon?

---

> ### Author Response · Authors · 2018-11-25
> **Authors' response**
>
> Thank you for providing thoughtful commentary and feedback on our submission.
>
> >>> Based on empirical evaluation, the paper cannot make any claim about the generality of the obtained results.
>
> We do not claim to have a fully general understanding of how these scaling phenomena will vary across arbitrary problems. However, one of our goals is to show that the scaling behavior of large-batch SGD differs significantly across problem domains, which is in contrast to the large body that explores these techniques almost exclusively in image classification ((You et al., (arxiv:1708.03888), (Goyal et al., arxiv:1706.02677), (Jia et al., arxiv:1807.11205)). The scaling behavior that we observed on these new problems quickly deviates from what we would expect to see on a standard image classification task.
>
> >>> [I]t is not clear how the definition of different training phases can help the practitioner to tune the training parameters
>
> Learning how to use these scaling observations to provide explicit guidance to practitioners given a particular problem configuration is an open problem. The goal of our work is to show that, in the presence of this speedup saturation phenomenon, simply optimizing within the existing SGD hyperparameter configuration space to accommodate a large batch size is not sufficient to enable significant reductions in training time or computational cost.
>
> >>> how are the empirical results obtained in the experiment section expected to depend on the specific dataset/benchmark?
>
> Right now, it does not appear as though there is a straightforward way to isolate the impact of a specific property like model architecture, because different datasets / problem domains require markedly different architectures. Even without this challenge, it is very difficult to isolate the exact effect of various dataset and model properties on convergence speed within the same problem domain, because these problems have complex relationships with properties of the overarching objective landscape.
>
> >>> what is a  batch size that does not allow one to 'fully utilize our available compute'?
>
> The goal of using large batch sizes is to ensure that GPU cycles, not communication bandwidth, is the bottleneck for overall throughput. By a batch size that does not allow one to ‘fully utilize available compute’, we are referring to a batch size that is small enough that communication bandwidth becomes the main bottleneck. Our main conclusion in this submission is that even though increasing batch size allows for the GPU cycles to be fully utilized, increasing batch size beyond a certain point no longer leads to proportional (or even any) improvement in overall time to convergence. In other words, the most cost-efficient batch size for a particular problem may still leave many GPUs sitting idle.
>
> >>> does the amount of over-parameterization in the model have any effects on the definition of the training phases?
>
> Yes; theoretical results link particular manifestations of over-parameterization to the presence of these training phases ((Ma et al., arxiv: 1712.06559), (Yin et al., arxiv:1706.05699)). However, in practice, it is difficult to isolate the effects of the over-parameterization itself, since changing the model architecture to increase the degree of over-parameterization changes the objective landscapes in ways that are difficult to characterize. Furthermore, different ways to change the amount of over-parameterzation (larger hidden layers, more hidden layers, etc.) may have different effects, and there is no clear way to choose a canonical over-paramerization method.

---

### Official Review · AnonReviewer1 · 2018-10-31
**Insightful empirical study of the effect of batch size for convergence speed**

**Rating:** 8
**Confidence:** 4

**Review:**

This paper empirically investigates the effect of batch size on the convergence speed of the mini-batch stochastic gradient descent of popular deep learning models. The fact that there is a diminishing return of batch size is not very surprising and there is a well-known theory behind it, but the theory doesn't exactly tell when we will start to suffer from the diminishing return. Therefore, it is quite valuable for the community to have an empirical analysis across popular ML tasks and models. In this regard, however, It would've been even nicer if the paper covered more variety of popular ML models such as Machine Translation, Speech Recognition, (Conditional) Image Generation, etc which open source implementations are readily available. Otherwise, experiments in this paper are pretty comprehensive. The only additional experiment I would be interested in is to tune learning rate for each batch size, rather than using a base learning rate everywhere, or simple rules such as LSR or SRSR. Since the theory only gives us asymptotic form of the optimal learning rate, empirically you should be tuning the learning rate for each batch size. And this is not totally unrealistic, because you can use a fraction of computational time to do cross-validation for searching the learning rate.

pros:
* findings provide us useful direction for future research (that data-parallelism centered distributed training is going to hit the limit soon)
* extensive experiments across 5 datasets and 6 neural network architectures

cons:
* experiments are a bit too much focused on image classification
* error bars in figures could've provided greater confidence in robustness of findings

---

> ### Author Response · Authors · 2018-11-25
> **Authors' response**
>
> Thank you very much for your review and comments.
>
> >>> It would've been even nicer if the paper covered more variety of popular ML models such as Machine Translation, Speech Recognition, (Conditional) Image Generation, etc which open source implementations are readily available.
>
> We hope to also verify these results across other interesting problem domains such as those you have suggested. For this paper, we selected image segmentation and language modeling tasks because these problems displayed markedly different convergence behavior than what has been hypothesized from existing empirical results, which primarily focus on image classification problems.
>
> >>> Since the theory only gives us asymptotic form of the optimal learning rate, empirically you should be tuning the learning rate for each batch size.
>
> Since the time of submission, we have performed additional experiments where we try out a variety of learning rates for each batch size, and produce a contour plot of the resulting training error at the end of training. We have included a link to the figure below [1]. We observe that as we increase batch size, it becomes more difficult to find a model with low training loss, regardless of the initial learning rate. We plan to include this figure as supplementary material in the appendix.

---

### Official Review · AnonReviewer3 · 2018-11-06
**Limited insights in the understanding of the batch size effect**

**Rating:** 5
**Confidence:** 3

**Review:**

The work presented relates to the impact batch-size on the learning performances of common neural network architectures.

Pro: having comprehensive study of the limit of gradient-based methods is very useful in practice. This work can help practitioner to limit the number of machines used for optimization.

Cons: very little can be deduced from these experiments:
- "Increasing the batch size beyond a certain point yields no improvement in wall-clock time to convergence, even for a system with perfect parallelism." was a know fact (they cite Ma et al (2017) who even proved it theoretically.
- "Increasing the batch size leads to a significant increase in generalization error, which cannot be mitigated by existing techniques.". It is not clear that all the regularization techniques have been tried by the authors, the increase of generalization error is very small, and there is no explanation or insight given by the authors to explain this phenomenon, making this finding of limited interest.
- "Dataset size is not the only factor determining the computational efficiency of large batch training." is something obvious to say, as there are plenty of factors that determine the computational efficiency (network connection, map-reduce implementation, etc.)

Even the suggestions for future work of the authors in the conclusion does not help much: they suggest to look at "alternative forms of parallelism", without citing or giving any clue of what could be such alternative forms.
Also, there is no discussion around lock-free

The authors refer to Ma et al. (2017) for a theoretical analysis of the effect of the batch size, but they skip all the past and very relevant literature on the topic of the effect of the batch size on the convergence. For example, it is recommended to increase the size of the batch size as the iterations increase.

Finally, there is no discussion on the lock-free gradient descent, that is often suggested as an alternative to batching.

In conclusion, I'm not convinced there is enough material to accept this paper at the next ICLR conference.

---

> ### Author Response · Authors · 2018-11-25
> **Authors' response**
>
> Thank you for your comments.  As you point out, Ma et al. (2017) have already shown that increasing the batch size indefinitely eventually stops yielding any improvement in convergence speed. Missing from this theoretical analysis is a prediction of what exact batch size is too large, rendering their results of limited use for practitioners. Although finding an optimal batch size a priori in the general case has proved elusive, our results demonstrate that this optimal/maximum batch size is heavily problem-dependent (compare the contour plots for DRN and ResNet34 in Figure 1), and they suggest, though do not prove, that current state-of-the-art results on image classification tasks are already nearing the maximum. To our knowledge, our work is the first large-scale empirical study of the effects of batch size on convergence speed and training loss. Current theoretical analyses fail to explore the saturation phenomenon as a function of dataset and model architecture; our results show that the optimal batch size is heavily dependent on these parameters.
>
> >>> It is not clear that all the regularization techniques have been tried by the authors, the increase of generalization error is very small, and there is no explanation or insight given by the authors to explain this phenomenon, making this finding of limited interest.
>
> This work primarily studies the effects of batch size on convergence speed and on the minimum achieved training loss; generalization error only compounds these problems. We have included an additional table of test errors, which demonstrates a significant increase in generalization error (e.g. 93.58% for test accuracy at batch size 64 vs. 86.93% at batch size 8k for ResNet34 on CIFAR-10). Other lines of work already explore the effect of batch size on generalization error (theoretical: Jastrzębski et al. (arxiv:1711.04623), Zhu et al. (arxiv:1803.00195), and experimental: You et al. (arxiv:1708.03888), Smith and Le (arxiv:1711.0048)), and so we do not make a significant study of it here.
>
> >>>  “Dataset size is not the only factor determining the computational efficiency of large batch training." is something obvious to say, as there are plenty of factors that determine the computational efficiency (network connection, map-reduce implementation, etc.)
>
> Our claim is poorly worded. Our intention is to claim that, contrary to Smith and Le (arxiv:1711.0048) and Goyal et al. (arxiv:1706.02677), increasing the dataset size does not yield a linear increase in the permissible batch size. In this submission, we support this claim in Figure 4, and we intend to bolster this claim further with future work on larger datasets.
>
> Our intention in this work is to understand the ability of large batch sizes to provide significant gains in training efficiency and speed. By measuring convergence speed in terms of training iterations rather than wall-clock time, we demonstrate that, regardless of the particular distributed implementations, these gains quickly become marginal or non-existent. A better wording of this claim would be that model architecture and other problem properties play a more decisive role in determining training efficiency than dataset size alone.
>
> >>> Suggestions for future work are not very defined/helpful and no alternate forms presented
> >>> No discussion on the lock-free gradient descent, that is often suggested as an alternative to batching
>
> Asynchronous and lock-free methods are exciting approaches that may help in the future to address the speedup saturation behavior we observe. In this work, we focus on synchronous mini-batch training because most recent large-scale training work focuses on this setting (Chen et al. (arxiv:1604.00981)).  A natural direction of future work would be to explore variations in model architecture on the same training data to help disentangle the effects of the dataset and model architecture on the amount of exploitable data parallelism in the problem.
>
> >>>  Skipped past other heuristics on increasing the size of the batch size as the iterations increase
>
> Jastrzębski et al. (arxiv:1711.04623) and Smith and Le (arxiv:1711.00489) show that increasing the batch size as training progresses has the same effect as decaying the learning rate, and Smith et al. (arXiv:1711.00489) support this claim empirically. However, in their paper they also show that there is a maximum batch size that they can scale to, which actually includes the geometric learning rate scaling that we studied. Therefore, the adaptive batch size also shows the same behavior as explained by the theoretical results of Jastrzębski et al. (arxiv:1711.04623).

---

### Public Comment · (anonymous) · 2018-11-13
**Not in SGD assumption regime.**

My concern about this paper is that most experiments are done in CIFAR10 and CIFAR10 sample size is ~50000.  The batch size they mainly discussed is > 8000.  In this regime, MB_size is comparable to total training samples.  Thus, SGD assumption MB_size << training sample does not hold.  The challenge can be directly due to gradient decent itself; not batch size effect in SGD.

The other issue: Fig.3 and Fig.4 SVHN are not consistent.  Could you please explain?

---

> ### Author Response · Authors · 2018-11-25
> **Authors' response**
>
> Thank you for your comments. We agree that the degradation in performance we observe is due to the increased batch size, though we argue in Figure 5 that the maximum admissible batch size is not sensitive to the dataset size. To provide a bit more context, the motivation behind our investigation is that there is a broad interest in increasing batch size for synchronous SGD in order to make better use of massively parallel hardware. It is well-known that, when using small batches, increasing the batch size yields a commensurate decrease in the number of iterations needed to converge, without adversely affecting the final training loss. However, for larger batch sizes, our findings show that this trend breaks down. More specifically, the point at which increasing the batch size no longer yields a decrease in the time to convergence depends sensitively on the type of data and the model architecture, but less so on the size of the dataset.
>
> Regarding the SVHN speedup curves: in Figure 3, we train on only 50k examples to compare datasets of the same size. In Figure 4, we are training on partitions of the full 600k example dataset.

---

### Author Response · Authors · 2018-11-25
**Authors' response**

We are very grateful to the reviewers for their time and for their thoughtful reading of the manuscript. We would like to make some general comments here:

- Existing theoretical results: Unfortunately, there is not yet a full theoretical treatment of the relationships between batch size, convergence speed, and best-achieved training loss. Some interesting theoretical results in this area exist (e.g. Ma et al. 2017, arxiv:1712.06559; Yin et al. 2018, arxiv:1706.05699), but they either rely on too-stringent assumptions, or they make claims that are too general to be of much use to practitioners. With this in mind, we see our more empirical contribution as an important step toward disentangling the effects of various problem parameters (e.g. model architecture, dataset size, learning rate) on the potential for data parallelism in synchronous SGD. In this way, we see our empirical results as complementary to existing theoretical work.

- Generality of our results: The nature of an empirical study of this sort is that results do not generalize to arbitrary other settings, and although we investigate our findings over a range of models, datasets, and learning rates, we make no claim that our results enjoy full generality. However, we hope these results will raise awareness among practitioners that the largest admissible batch size depends heavily on these parameters, and that they must be careful not to increase the batch size too much, lest they increase their GPU utilization without reducing the number of iterations to convergence.

- Effect on generalization gap: This work primarily studies the effects of batch size on convergence speed and on the minimum achieved training loss; generalization error only compounds these problems, and we do not study it extensively. Our results confirm the existence of the generalization gap across several problem domains (see Figure 5 in the appendix). We also show that even with the LR scaling rule, the generalization gap persists, especially for non-image-classification problems.

---

### Meta-Review · Area_Chair1 · 2018-12-14
**Interestng empirical analysis but insights might be limited**

**Confidence:** 3
**Recommendation:** Reject

**Metareview:**

The paper presents an interesting empirical analysis showing that increasing the batch size beyond a certain point yields no decrease in time to convergence. This is an interesting finding, since it indicates that parallelisation approaches might have their limits. On the other hand, the study does not allow the practitioners to tune their hyperparamters since the optimal batch size is dependent on the model architecture and the dataset. Furthermore, as also pointed out in an anonymous comment, the batch size is VERY large compared to the size of the benchmark sets. Therefore, it would be nice to see if the observation carries over to large-scale data sets, where the number of samples in the mini-batch is still small compared to the total number of samples.